# Generation and characterization of a *Meflin*-CreER^T2 transgenic line for lineage tracing in white adipose tissue

Takahide Kuwano[1], Hironori Izumi[2], Muhammad Rahil Aslam [1], Yoshiko Igarashi[1], Muhammad Bilal[1], Ayumi Nishimura[1], Yoshiyuki Watanabe[1], Allah Nawaz[1], Tomonobu Kado[1], Koichi Ikuta[3], Seiji Yamamoto[4], Masakiyo Sasahara[4], Shiho Fujisaka[1], Kunimasa Yagi[1], Hisashi Mori[2], Kazuyuki Tobe[1] *

**1** First Department of Internal Medicine, University of Toyama, Toyama-shi, Toyama, Japan, **2** Department of Molecular Neuroscience, University of Toyama, Toyama-shi, Toyama, Japan, **3** Department of Virus Research, Laboratory of Immune Regulation, Institute of Frontier Life and Medical Sciences, Kyoto University, Sakyo-ku, Kyoto, Japan, **4** Department of Pathology, University of Toyama, Toyama-shi, Toyama, Japan

* tobe@med.u-toyama.ac.jp

**Data Availability Statement:** All relevant data are within the manuscript and its Supporting Information files.

## Abstract

*Meflin* (*Islr*) expression has gained attention as a marker for mesenchymal stem cells, but its function remains largely unexplored. Here, we report the generation of *Meflin-CreER^T2* mice with *CreER^T2* inserted under the *Meflin* gene promoter to label *Meflin*-expressing cells genetically, thereby enabling their lineages to be traced. We found that in adult mice, *Meflin*-expressing lineage cells were present in adipose tissue stroma and had differentiated into mature adipocytes. These cells constituted Crown-like structures in the adipose tissue of mice after high-fat diet loading. Cold stimulation led to the differentiation of *Meflin*-expressing lineage cells into beige adipocytes. Thus, the *Meflin-CreER^T2* mouse line is a useful new tool for visualizing and tracking the lineage of *Meflin*-expressing cells.

## Introduction

White adipose tissue (WAT) is responsible for the storage and release of energy in response to environmental changes. Excess energy from excessive diets has led to obesity and a rapid increase in the number of people suffering from diabetes and other diseases [1,2]. Understanding the development of adipose tissue and its responses to environmental changes is important. Adipose tissue grows in two manners: the adipocyte cells themselves grow in size, and the number of adipocytes increases. Increases in the number of adipocytes can occur through the clonal proliferation of mature adipocytes and the differentiation of preadipocytes, which are the precursor cells of adipocytes [3]. The mechanism of differentiation from mesenchymal stem cells, which are thought to be more immature than preadipocytes, to the adipocyte lineage is still unclear.

*Meflin* (*Islr*) expression has received much attention as a marker for mesenchymal stem cells. Meflin has a leucine-rich repeat and immunoglobulin sequence and is located on the cell

**Funding:** This work was supported by JSPS KAKENHI Grants-in-Aid for Scientific Research(B) (20H03730; https://www.jsps.go.jp/j-grantsinaid/index.html), Firstbank of toyama scholarship foundation reserch grant (https://www.first-bank.co.jp/outline/local), the Mitsubishi foundation (201910031:https://www.mitsubishi-zaidan.jp), the Naito Foundation (https://www.naito-f.or.jp/jp/index.php),Grant from Japan Diabetes Foundation (http://www.j-df.or.jp), the Uehara Memorial Foundation (https://www.ueharazaidan.or.jp) to KT. The funders had no role in study design, data collection and analysis, decision to publish, or preparation of the manuscript.

**Competing interests:** The authors have declared that no competing interests exist.

membrane at a glycosylphosphatidylinositol anchor [4]. During early development (E10), *Meflin* is not expressed in the nervous system and is only expressed in the mesenchymal tissue of the head and trunk [5]. Meflin (mesenchymal stromal cell- and fibroblast-expressing Linx paralogue) was in fact named for its potential usefulness as a marker of mesenchymal stem cells [4].

In recent years, the gene expression patterns of preadipocytes and more immature cells in adipose tissue have been analyzed and clarified at the single cell level. Single-cell RNA sequencing analyses of the stromal vascular fraction (SVF) in multiple types of WAT have shown that the common features of cell populations expressing *Meflin* are an undifferentiated state, high levels of proliferation markers, and the presence of stromal cell characteristics [6–8]. However, the involvement of *Meflin* in adipose tissues has not been studied.

We generated *Meflin-CreER^T2* mice under the assumption that *Meflin* might play an important role in the development and formation of adipose tissue. Tamoxifen induces Cre enzymes at any point in time, enabling genetic modification. Lineage tracing can therefore be performed by crossing these mice with a reporter mouse. We analyzed how mesenchymal stem cells and their progeny differentiate into cells in adipose tissue in adults. In vivo, *Meflin*-expressing lineage cells differentiated into mature adipocytes and constituted Crown-like structures (CLS). In addition, *Meflin*-expressing lineage cells were found in multivesicular, brownish-toned cells, suggesting their differentiation into cells that were thought to be beige adipocytes.

## Materials and methods

### Ethics statement

The use of animals was approved by the University of Toyama's Animal Laboratory Facility and Use Committee. All the experiments were performed in accordance with the relevant guidelines and regulations and were approved by the Committee for Institutional Animal Care and Use of the University of Toyama, Toyama, Japan (License No. A2018med-355/A2018MED-52).

### Mice

Mice were given free access to food and water and were kept under standard laboratory conditions. *C57BL/6J* wild-type mice and ROSA^mTmG mice (Jax 007676) [9] were purchased from The Jackson Laboratory (USA). Rosa26-CAG-lox-stop-lox-tdTomato (Rosa26-tdTomato) mice [10] were provided by Dr. Kouichi Ikuta (Kyoto University). Genotyping of the purchased mouse strains was performed using the PCR primers and protocols provided by the supplier. *Meflin-CreER^T2* mice were crossed with a Cre-dependent reporter line. Mice were fed a standard rodent chow at ambient temperature (24°C) under a 12-hour light-dark cycle. *Meflin-CreER^T2*/Rosa26-tdTomato mice and *Meflin-CreER^T2*/ROSA^mTmG mice were fed a high-fat diet (Research Diets, Cat# D12492) for eight weeks. For cold stimulation, tamoxifen-administered mice were placed in conventional cages and then subjected to 6°C for 48 h using a rodent incubator [11].

### Construction of BAC-Tg Meflin-CreERT2

The plasmid pCAG-CreER^T2 [12] was obtained from Addgene (Cat# 14797). We constructed a *CreER^T2* DNA fragment containing 5' and 3' homology arms derived from the *Meflin* genomic sequence using high-fidelity PCR and recombinant DNA methods. The resulting DNA fragment was inserted at the translational initiation Met of the mouse *Meflin* gene in a bacterial

artificial chromosome (BAC) genomic clone (B6Ng01-165L10) using high-efficiency counter-selection recombineering [13] to yield pTg-*Meflin-CreER^T2*.

## Generation of BAC Meflin-CreERT2 transgenic mouse strains

Purified pTg-*Meflin-CreER^T2* BAC DNA was microinjected into the pronuclei of fertilized one-cell embryos from C57BL/6 mice. Founder mice were crossed with C57BL/6 mice to produce +/*Meflin-CreER^T2* mice. For the genotyping of the Tg mouse lines using a Southern blot analysis, genomic DNA prepared from a tail biopsy was digested with SphI, separated by electrophoresis on a 0.8% agarose gel, and transferred to a nylon membrane Hybond-N$^+$(Cytiva, Cat# RPN2222B). Hybridization was conducted with a 1286-bp $^{32}$P-labeled DNA fragment derived from the open reading frame of *Meflin*. The detected band sizes in endogenous and transgenic genes were 2.9 kb and 4.9 kb, respectively. Semi-quantification of bands in Southern blot was conducted using ImageJ software. An image file was imported, and the color tone was inverted. An area of the bund was measured in the form of a square and its average intensity was measured. The average intensity value of the measured the band was subtracted by the background value and used for comparison.

## Genotyping

Routine genotyping of *Meflin-CreER^T2* mice was performed using PCR. Mouse tails were biopsied, and genomic DNA was extracted. The PCR enzyme was Tks Gflex DNA Polymerase (TaKaRa, Cat# R060A). The sequence of the forward primer was CCAGCTAAACATGCTT CATCGT, and the sequence of the reverse primer was TCGCTCGACCAGTTTAGTTACC. These primers yielded a PCR product of 391 bp in the *Meflin-CreER^T2* allele. The PCR conditions (Touchdown PCR) [14] were set as follows: 1 cycle of 2 min at 94˚C; 10 cycles of 20 s at 94˚C, 30 s at 65˚C (reduced by -0.5˚C per cycle), and 30 s at 68˚C; 28 cycles of 15 s at 94˚C, 15 s at 60˚C, and 30 s at 72˚C; and 1 cycle of 2 min at 72˚C.

## Tamoxifen administration

Tamoxifen (Sigma-Aldrich, Cat# T5648) was dissolved in sunflower seed oil (Wako, Cat# 196–15265), shaken and stirred overnight at 37˚C, and stored at 4˚C in a shaded environment (22.5 mg/mL). Tamoxifen was administered orally to 6-week-old mice at a dosage of 0.225 mg/g body weight per day for 5 consecutive days [15]. Tissues were collected 1 day, 4 weeks, or 8 weeks after tamoxifen administration.

## Immunohistochemistry

Tissues were soaked in 4% paraformaldehyde at 4˚C and fixed overnight. The samples were then soaked in 30% sucrose/PBS and stored overnight at 4˚C. Finally, the samples were embedded in OCT compound (Sakura Finetek, Cat# 45833) and frozen at -80˚C in a deep freezer. The frozen blocks were sliced to 10–20 μm on a cryostat (CM3050S, Leica) [16]. WAT was fixed in 4% paraformaldehyde and embedded in paraffin wax. Paraffin-embedded blocks were sectioned to 5-μm using a microtome (REM710, Yamato). Immunohistochemistry was performed on frozen sections or paraffin sections using standard protocols. Primary antibodies were applied to the sections and incubated overnight at 4˚C. Secondary antibodies were used against the host animals corresponding to the primary antibody. The primary antibodies were goat polyclonal anti-tdTomato antibody (1:100; ORG Origene Technologies, Cat# AB8181-200), rabbit polyclonal anti-UCP-1 antibody (1:500; Abcam, Cat# ab10983), rabbit polyclonal anti-Perilipin antibody (1:100; Santa Cruz Biotechnology, Cat# sc-67164), and rat monoclonal

anti-F4/80 FITC conjugated antibody (1:500; Abcam, Cat# ab60343). The secondary antibodies were Alexa Fluor-488 donkey anti-rabbit (1:500, Life Technologies, Cat# A-21206) and Alexa Fluor-594 donkey anti-goat (1:500, Life Technologies, Cat# A-11058). The stained slices were mounted with DAPI Fluoromount-G (SBA Southern Biotechnology, Cat# 0100–20). AlexaFluor 488 phalloidin (Invitrogen, Cat# R37110) was used to stain for F-actin and to visualize the cytoskeleton in the frozen sections. The sections were imaged using an Olympus BX61/DP70 and a Zeiss LSM780 confocal microscope.

## Wholemount and confocal imaging

Adipose depots were dissected and cut into $0.5 \times 0.5 \times 0.2$-cm blocks. These blocks were mounted onto slides with Fluoromount (Diagnostic BioSystems, Cat# K024) and imaged using a Zeiss LSM780 confocal microscope [17]. Brightness and contrast were adjusted to improve the visibility of the cell membranes, and EGFP+/tdTomato+ adipocytes were quantified using ImageJ software [18].

## Glucose tolerance test and insulin tolerance test

Oral glucose tolerance test (2 g/kg body weight) and intraperitoneal insulin tolerance test (1.0 unit/kg body weight) were performed after 4-hour and 2-hour fasting period, respectively. Blood samples were taken from the tail vein at specific times and glucose levels were measured using STAT STRIP Express 900 (Nova Biomedical, Waltham MA).

## Quantitative real-time PCR

Total RNA was extracted with RNeasy mini kit (QIAGEN, Cat# 74106), and complementary DNA was synthesized with PrimeScript RT Master Mix (Takara, Cat# RR036A), in accordance with the manufacturer's protocol. Quantitative PCR of the genes was performed using TB Green Fast qPCR Mix (Takara, Cat# RR430S). Fluorescence was analyzed in the Mx3000P QPCR System (Agilent Technologies, Santa Clara, CA, USA). The thermal cycling conditions were 1 cycle at 95˚C for 30 s, followed by 45 cycles of 10 s at 95˚C and 30 s at 60˚C. Each sample was run in duplicate. The relative mRNA levels were normalized to *Tbp* mRNA. The relative mRNA expression levels were calculated using the $2^{\Delta\Delta CT}$ method. Primer sequences are listed in S1 Table.

## Statistical analysis

Statistical analysis was performed with JMP® 14 (SAS Institute Inc., Cary, NC, USA). The statistical significance of between-group differences was evaluated by using unpaired two-tailed Student's t-tests. Differences among more than two groups were evaluated for statistical significance by ANOVA with the Tukey-Kramer HSD post hoc test for multiple comparisons. A P value of less than 0.05 was considered significant. Data are presented as the mean ± SEM.

## Result

### Generation of *Meflin-CreER^T2* mice

*Meflin-CreER^T2* transgenic mice were generated to enable the lineage tracing of mesenchymal stem cells and adipocyte progenitors. The *CreER^T2* coding sequence was inserted downstream of the start codon of exon 2 of the *Meflin* gene (Fig 1A). *Meflin-CreER^T2* BAC DNA was then microinjected into the pronuclei of fertilized one-cell embryos from C57BL/6 mice. Forty-three offspring mice were obtained. A Southern blot analysis confirmed that the four founder mice carried the *CreER^T2* allele. The transgenic and wild-type alleles were identified at 4.9 and

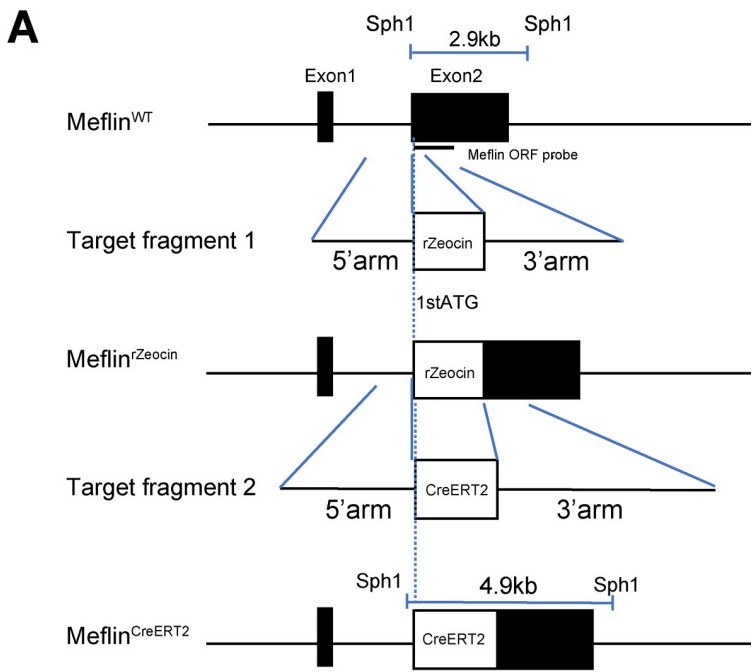

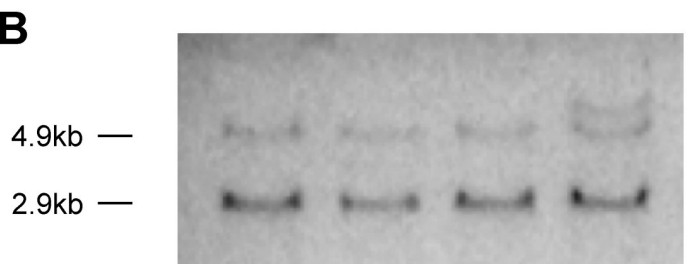

**Fig 1. Generation Meflin-CreER[T2] mice.** a) First, BAC carrying the *Meflin* gene was introduced together with a zeocin resistance gene (*rZeocin*) downstream of the translation start point Met. Next, counter-selection using a *CreER[T2]* fragment with the 5' and 3' homologous arms of the *Meflin* gene was performed. b) Southern blot analysis. Genomic DNA was digested with Sph1 and detected using a *Meflin*-ORF probe. The transgenic allele was identified at 4.9 kb, and the wild-type allele was identified at 2.9 kb.

2.9 kb, respectively. A semi-quantitative analysis by ImageJ showed that an intensity of the 4.9 kb band was half that of the 2.9kb band, indicating that one copy of the transgene allele was inserted (Fig 1B). The heterozygous *Meflin-CreER[T2]* mice were viable, fertile, and behaved similarly to wild-type mice. Meflin gene expression showed no difference in the adipose tissue in the presence or absence of the transgenic allele (S1A Fig).

## Functional *Meflin*-CreER[T2] activity in vivo

To evaluate the functional expression of *CreER[T2]* in vivo throughout the whole body, *Meflin-CreER[T2]* mice were crossed with *Rosa26-tdTomato* reporter mice containing the *tdTomato* gene, the expression of which requires the deletion of loxP-flanked stop sequences [10].

The *Meflin-CreER[T2]/Rosa26-tdTomato* transgenic mice were orally treated with tamoxifen at 6 weeks of age for 5 consecutive days. Four weeks after the treatment, the mice were sacrificed, and WAT and several other types of tissues were harvested.

In WAT, tdTomato expression was observed in mature adipocytes and brown adipocytes (Fig 2A and 2B). In skeletal muscle tissue, tdTomato expression was observed in cells located at the periphery of muscle fibers (Fig 2C). In mammary glands, tdTomato fluorescence was found in star-shaped cells surrounding the mammary ducts (Fig 2D). In the liver, fluorescence was observed in perivascular stroma that was in close proximity to a vein (Fig 2E). In the pancreas, some of the glandular cells were found to have a strong fluorescent signal (Fig 2F). In the intestine, the tdTomato signal was found throughout the cytoplasm of a portion of the monolayer columnar epithelium (Fig 2G). In the testis, expression was found in Leydig cells located in the interstitium (Fig 2H). In the kidneys, tdTomato was expressed in cells comprising the glomeruli. It was also expressed in cells with foot processes in the peritubular stroma (Fig 2I). The specificity and efficiency of recombination did not differ between male and female mice. These results revealed the recombination of *CreER^{T2}* throughout the body and the distribution of various *Meflin*-expressing lineage cells.

## Lineage tracing of white adipose tissue

To evaluate the distribution of *Meflin*-expressing lineage cells in WAT and the differentiation of preadipocytes into mature adipocytes, we analyzed WAT from *Meflin-CreER^{T2}/Rosa26-td-Tomato* mice fed a normal diet for 4 weeks or a high-fat diet for 8 weeks. The expression of mesenchymal stem cell (MSC) markers (including *Meflin*, *Cd105*, *Sca-1*) was increased in the high-fat diet-loaded mice compared to the normal diet-loaded mice (S1B Fig).

After 4 weeks of the normal diet, several *Meflin*-expressing lineage cells were located in the stroma between mature adipocytes. A number of cells with tdTomato fluorescence were present in the membranous structures surrounding gonadal WAT (gWAT). *Meflin*-expressing lineage cells mainly existed in the capsule areas surrounding the periphery of gWAT and moved from the capsule area to the internal adipose tissue. We also observed the co-staining of fluorescent signals from tdTomato and perilipin, a marker of mature adipocytes (Fig 3A and 3B).

Inflammatory markers were elevated in mice fed a high-fat diet for 8 weeks, compared with mice fed a normal diet (S1D Fig). Transgenic expression of *Meflin-CreER*^{T2} did not affect glucose tolerance (S1E and S1F Fig). After 8 weeks of HFD treatment, tdTomato-positive and perilipin-negative cells from *Meflin*-expressing lineages had formed ring-shaped structures and multinucleated giant cells (Fig 3C). These ring-like structures are known as CLS. CLS are characteristic structures in which macrophages and multinucleated giant cells surround and process adipocytes that are subjected to cell death during the obesity process [19]. We confirmed that CLS in WAT co-expressed tdTomato and F4/80 (S1G Fig). Small multicellular adipocytes were also seen, and *Meflin*-positive cells were located adjacent to these cells (Fig 3D). To assess the contribution of *Meflin*-expressing lineage cells to mature white adipocytes, *Meflin-CreERT2*/ROSA^{mTmG} mice were subjected to a high-fat diet for 8 weeks to stimulate differentiation from progenitor adipocytes to mature adipocytes. The normal diet-fed mice had an EGFP-positive rate of 26.7% and a tdTomato-positive rate of 73.3% (Fig 3E and 3G). In high-fat diet Mature adipocytes expressing EGFP and tdTomato-expressing cells accounted for 50.4% and 49.6% of the cell population, respectively (Fig 3F and 3G). These results suggest that in adult adipose tissues in animals fed an HFD, Meflin lineage cells constitute the CLS. However, whether these cells are adipocytes, macrophages or preadipocytes remains unknown.

## *Meflin* lineage cells become beige like adipocytes

As shown in Fig 3D, we observed small, multi-spore-like adipocytes that were morphologically similar to beige adipocytes [20]. Beige adipocytes are brown-like adipocytes that have a

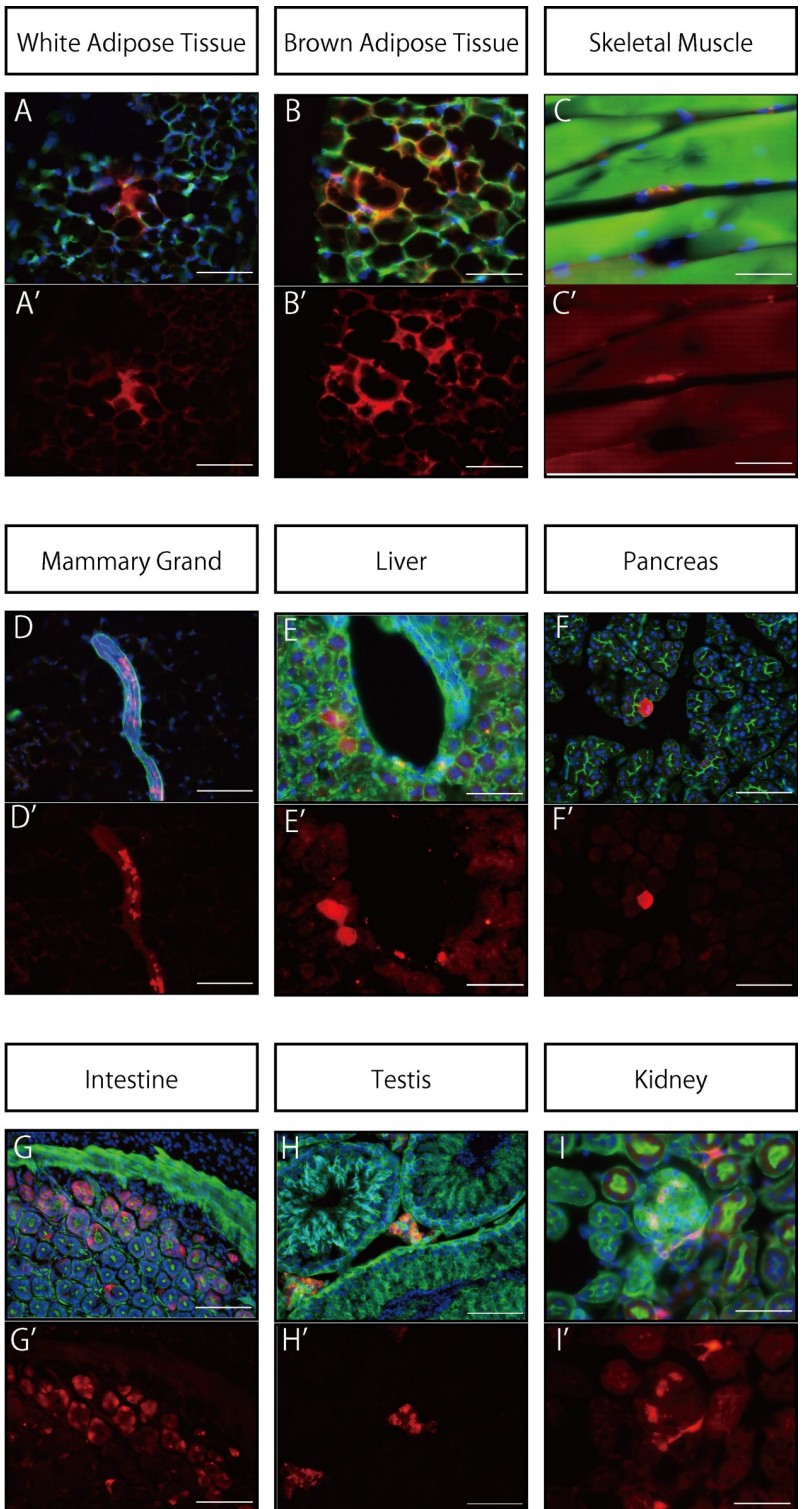

**Fig 2. Expression distribution of *Meflin* lineage cells in vivo.** Organ tissues from *Meflin*-CreER$^{T2}$/Rosa26-tdTomato mice treated with tamoxifen beginning at 6 weeks of age and fed a normal diet for 4 weeks were analyzed histologically. Nuclei were stained using DAPI (blue); staining with the F-actin marker phalloidin (green) was used to define the cytoskeleton. The tdTomato signal (red) was found in gonadal white (A) and brown (B) adipocytes. In skeletal muscle (C), mammary ducts (D), liver (E), testes (H), and kidneys (I), tdTomato expression was found in cells located in the interstitium. In the pancreas, expression was observed in glandular cells (F); expression in the kidneys was seen in cells

constituting the glomerulus (I). The scale bars represent 50 μm in (B), (B'), (C), (C'), (E), (E'), (I) and (I') and 100 μm in all the other panels. A-I: merged (tdTomato + F-Actin + DAPI), A'-I': tdTomato.

multilocular morphology and express UCP1 but are found in WAT [21]. Since they are induced in WAT exposed to cold or other inducers [21], cold stimulation experiments were performed to further investigate the involvement of *Meflin*-expressing lineage cells in beige adipocytes. *Meflin-CreER^T2/Rosa26-tdTomato* mice were subjected to cold stimulation at 6°C for 48 hours and inguinal WAT (iWAT) was harvested. Cold stimulation did not affect the expression of MSC marker genes including *Meflin*, *Cd105*, *Sca-1* (S1C Fig). To determine the cellular characteristics, we stained iWAT with anti-UCP1 and anti-tdTomato antibodies and observed that cells with a beige-like adipocyte morphology were co-stained (Fig 4A, 4A', 4B and 4B'). The results suggest that cold stimulation led to the differentiation of *Meflin*-expressing lineage cells into beige adipocytes.

## Discussion

We generated *Meflin-CreER^T2* mice and traced the distribution of *Meflin* in both systemic tissues and WAT. We confirmed the occurrence of recombination with *CreER^T2* throughout the body and observed that many lineage cells were located in the stroma.

*De novo* adipogenesis in gWAT reportedly occurs in fibroblasts in an outer structure, known as the capsule, that surrounds adipose tissue [22]. WT1 lineage mesothelial cells have been reported as the origin of de novo adipogenesis in gWAT [23]. Our present findings were similar to these previous studies. *Meflin*-expressing lineage cells mainly exist in capsule areas, and the cells move from the capsule area to the internal adipose tissue. These observations suggest that *Meflin*-positive cells in the fibrous stroma surrounding gWAT generate new adipocytes as they migrate from outside the membrane to the inside.

We found that in mice fed a high-fat diet for 8 weeks, the CLS were composed of *Meflin*-expressing lineage cells in WAT. CD68 (a common macrophage marker)-positive and CD34-positive cells reportedly form a ring structure in WAT and are involved in adipocyte differentiation and proliferation [24]. Our results suggest that *Meflin*-expressing lineage cells constitute the CLS and are involved in the remodeling of WAT.

The origin of beige adipocytes is still unclear, but they are reportedly derived from the conversion of existing white adipocytes [25,26]. In other words, beige adipocytes are thought to differentiate from progenitor cells in white adipocytes. PDGFRa-positive cells in iWAT are thought to be the precursors of beige adipocytes [27]. We observed the presence of morphologically similar beige-like cells in iWAT after cold stimulation that were positive for both UCP1 and tdTomato. These findings suggest that *Meflin* is expressed in precursors, such as PDGFRa, that can differentiate into beige adipocytes.

Single cell RNA sequencing is a recent technological development that has made it possible to classify a previously unexplored population of SVFs in WAT, known as SVF progenitor adipocytes or adipose stem cells, based on the type and degree of gene expression [6–8]. In the present report, we observed the differentiation of *Meflin*-expressing lineage cells into mature adipocytes by the expression of *CreER^T2* in adipose tissue during the adult phase, confirming its usefulness as a marker for adipose stem cells.

A limitation of this study is that the induction of *CreER^T2* was limited to the adult phase. Future evaluation of adipose tissue during early development and the neonatal periods will be necessary to perform detailed studies on adipose tissue development and the lineage tracing of *Meflin*-expressing lineage cells.

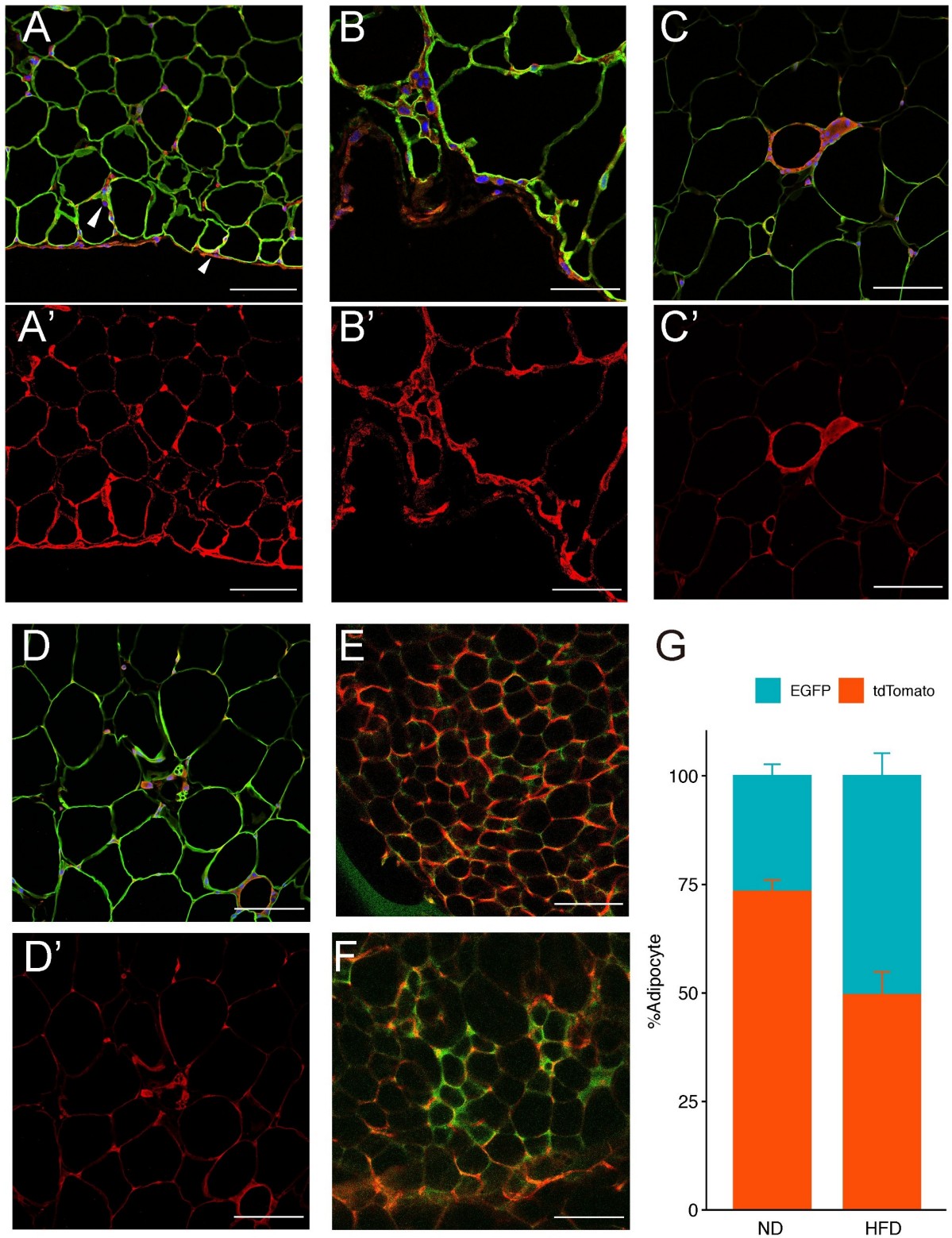

**Fig 3. Adipose tissue lineage tracing.** *Meflin*-CreER<sup>T2</sup>/Rosa26-tdTomato mice were fed a normal diet for 4 weeks (ND 4w) or a high-fat diet for 8 weeks (HFD 8w) and gonadal WAT (gWAT) samples were examined. WAT was stained with the adipocyte markers perilipin (green),

tdTomato (red), and DAPI (blue). (A) ND4w gWAT is surrounded by membranous structures with tdTomato signals. *Meflin*-expressing lineage cells appear to have penetrated into the adipose tissue stroma (arrowhead). (B) Strong magnified image of membranous structure in ND4w gWAT. (C) HFD8w gWAT. *Meflin*-expressing lineage cells comprise Crown-like structure, a characteristic structure found in adipose tissue. (D) HFD8w gWAT. Small, multiporous adipocytes are visible, and tdTomato-positive cells can be seen adjacent to the cells. MeflinCreER^T2/ROSA^mTmG mice were loaded with a normal diet (E) and a high-fat diet (F) for 8 weeks. Whole-mounted gWAT samples were then examined. Confocal laser microscopy was used to image tdTomato-positive and EGFP-positive adipocytes. Each fluorescent cell was counted (G). Scale bars represent 50 μm in (B) and (B') and 100 μm in all the other panels. A-D: Merged (tdTomato + Perilipin + DAPI), A'-D': tdTomato, E-F: EGFP + tdTomato.

A single cell RNA sequencing analysis in humans reported a positive effect on blood glucose when the number of retained pre-beige adipocytes, known as VPMs, was relatively high [28]. We found that *Meflin*-expressing lineage cells can become beige-like cells. Consequently, we would like to evaluate the effects of Meflin-positive cells and their lineage cells on adipose tissue and glucose metabolism in the future.

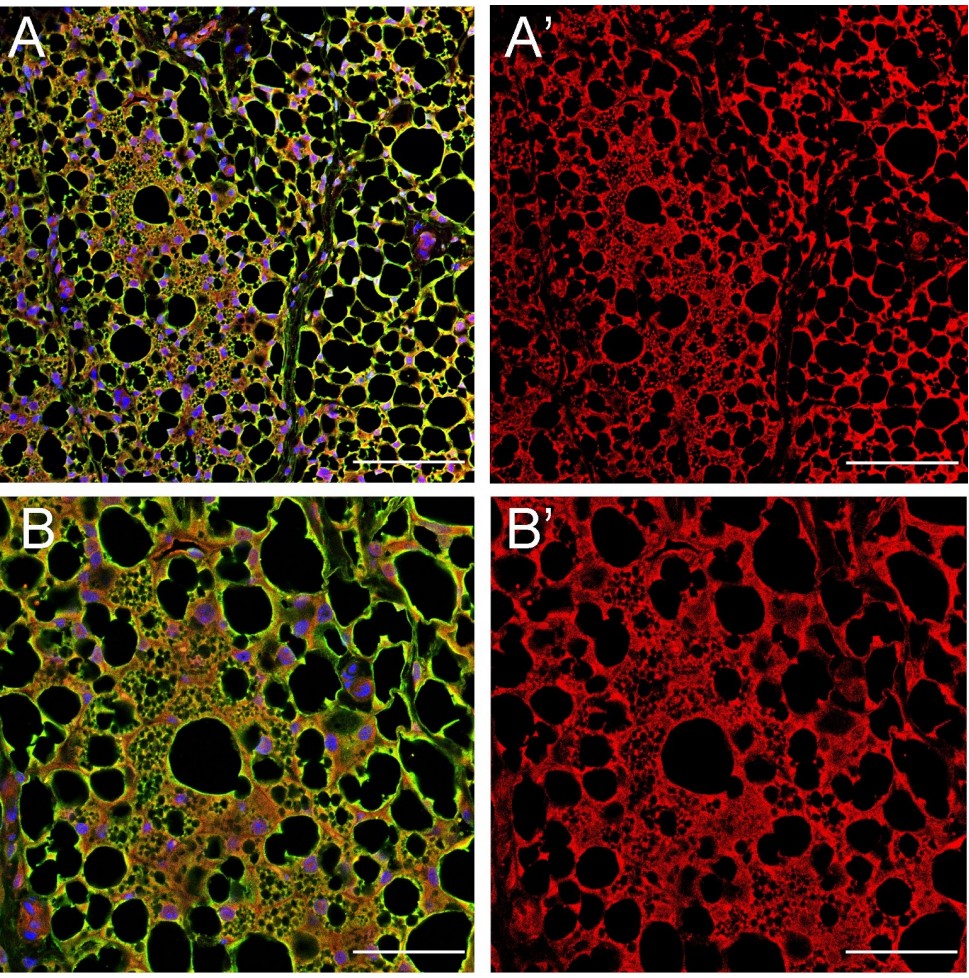

**Fig 4. *Meflin* lineage cells become beige like adipocytes.** *Meflin*-CreER^T2/Rosa26-tdTomato mice were subjected to cold stimulation at 6°C for 48 h. Inguinal WAT (iWAT) was then harvested. (A, A') Immunostaining image shows tdTomato (red)- and UCP1 (green)-positive cell masses. (B, B') Magnified images of A and A'. The cells in the center have the morphological characteristics of beige adipocytes. Scale bars represent 100 μm in (A) and (A') and 50 μm in (B) and (B'). A, B: Merged (UCP1 + tdTomato + DAPI), A', B': tdTomato.

## Supporting information

**S1 Fig.** A. Meflin mRNA levels in the gonadal WAT (n = 4–5). B. QPCR of mesenchymal stem cell related genes in gonadal white adipose tissues of mice treated with normal diet (ND) or high-fat diet (HFD) for 8weeks (n = 4–5). *p < 0.05 by unpaired, 2-tailed t test. C. QPCR of mesenchymal stem cell related genes in inguinal white adipose tissues of mice treated with room temperature (RT) or cold stimulation (Cold) (n = 3–4). D. QPCR of inflammation related genes in gonadal white adipose tissues of mice treated with normal diet (ND) or high-fat diet (HFD) for 8weeks (n = 4–5). E. Oral glucose tolerance test (OGTT) of HFD-fed mice for 8 weeks with the transgenic allele (n = 6) or WT allele (n = 5). F. Insulin tolerance test (ITT) of HFD-fed mice for 8 weeks with the transgenic allele (n = 3) or WT allele (n = 4). G. The Crown like structure were consisted of the cells expressing tdTomato and F4/80. *Meflin*-CreER^T2/Rosa26-tdTomato mice were fed a high-fat diet for 8 weeks and gonadal WAT (gWAT) samples were examined. *Meflin* lineage cells expressed tdTomato (red) and F4/80 (green). F4/80 is a mature macrophage marker. Nuclei were stained by DAPI (blue). Scale bars represent 50 μm.
(TIF)

**S1 Table. Primers used for qPCR analyses.**
(PDF)

**S1 Raw images.**
(PDF)

## Acknowledgments

The plasmid pCAG-CreERT2 was a gift from Dr. Connie Cepko (Addgene plasmid # 14797). Mouse BAC clone (B6Ng01-165L10P) derived from C57BL/6N strain was kindly provided from Riken bio-resource center (Riken BRC, Tsukuba). The pABRG and pR6K-photo-rspL-bsd plasmids used for BAC recombination were provided by Dr. A Francis Stewart.

## Author Contributions

**Conceptualization:** Takahide Kuwano, Hisashi Mori, Kazuyuki Tobe.

**Data curation:** Takahide Kuwano.

**Formal analysis:** Takahide Kuwano.

**Funding acquisition:** Kazuyuki Tobe.

**Investigation:** Takahide Kuwano, Hironori Izumi, Muhammad Rahil Aslam, Hisashi Mori.

**Methodology:** Takahide Kuwano, Hironori Izumi, Hisashi Mori.

**Project administration:** Kazuyuki Tobe.

**Resources:** Koichi Ikuta, Seiji Yamamoto, Masakiyo Sasahara, Hisashi Mori.

**Supervision:** Kazuyuki Tobe.

**Visualization:** Takahide Kuwano.

**Writing – original draft:** Takahide Kuwano.

**Writing – review & editing:** Yoshiko Igarashi, Muhammad Bilal, Ayumi Nishimura, Yoshiyuki Watanabe, Allah Nawaz, Tomonobu Kado, Shiho Fujisaka, Kunimasa Yagi, Hisashi Mori, Kazuyuki Tobe.

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
