## [Decision Letter · Decision Letter 0]

30 Dec 2020

PONE-D-20-35143

Generation and characterization of a Meflin-CreERT2 transgenic line for lineage tracing in white adipose tissue

PLOS ONE

Dear Dr. Tobe,

Thank you for submitting your manuscript to PLOS ONE. After careful consideration, we feel that it has merit but does not fully meet PLOS ONE’s publication criteria as it currently stands. Therefore, we invite you to submit a revised version of the manuscript that addresses the points raised during the review process.

We look forward to receiving your revised manuscript.

Kind regards,

Nobuyuki Takahashi, Ph.D.

Academic Editor

PLOS ONE

Reviewers' comments:

Reviewer's Responses to Questions

**Comments to the Author**

1. Is the manuscript technically sound, and do the data support the conclusions?

Reviewer #1: Yes

Reviewer #2: Yes

2. Has the statistical analysis been performed appropriately and rigorously? 

Reviewer #1: Yes

Reviewer #2: N/A

3. Have the authors made all data underlying the findings in their manuscript fully available?

Reviewer #1: Yes

Reviewer #2: Yes

4. Is the manuscript presented in an intelligible fashion and written in standard English?

Reviewer #1: Yes

Reviewer #2: Yes

5. Review Comments to the Author

Reviewer #1: The authors generated the tamoxifen-induced transgenic mice with Meflin, which w

as a major maker of the mesenchymal stem cell. Meflin was expressed on the crown

-like structure (CLS) in the adipose tissue of high fat diet-fed mice. The Mefli

n-expressing lineage cells were differentiated to the beige adipocytes by the co

ld exposure.

1. To what degree did Mefrin mRNA and protein expression levels increase in the

adipose tissue of Meflin transgenic mice compared to those of control mice?

2. Do other mesenchymal stem cell (MSC) markers, such as CD90, CD105 and Sca-1 e

levate in the adipose tissues of high fat diet-fed mice or after the cold exposu

re?

3. Please show the ratio of Meflin-positive cells in the adipose tissue of norma

l chow-fed mice in Figure 3F.

4. It is recommended that the authors measure the expression levels of Meflin in

UCP-1 positive cells of the adipose tissue after cold stimulation.

5. There was no description of Figure 4B’ in Result section.

6. The phenotypes of high-fat diet-fed Meflin transgenic mice, such as inflammat

ion and insulin sensitivity, are not mentioned in the text. Please specify those

phenotypes.

Reviewer #2: In this article, the authors generated Meflin-CreERT2 mice, and they performed lineage tracing using this mouse model. They showed that Meflin-expressing lineage cells exists in various tissues. In WAT, Meflin-expressing lineage cells were present in stroma and differentiated into mature adipocytes under HFD feeding condition. They also showed that Meflin-expressing lineage cells can differentiate into beige adipocytes under cold temperature. This article contains interesting findings. Please respond to the concerns listed below.

1. L120, “2 min at 4 ℃” should be “2 min at 94 ℃”.

2. The authors should determine the copy number of inserted DNA in mice used this study (Fig.2 ~) by Southern blotting.

3. Fig.2. tdTomato-derived fluorescence in some images is very weak and blurry, and it is very difficult to judge which cells are tdTomato positive cells. Especially, the fluorescence from brown adipose tissue, liver, intestine, testis, and kidney looks very weak, and it is difficult to exclude the possibility that these fluorescences are autofluorescence. The authors should show clearer images and clarify that these fluorescencea are not from autofluorescence.

4. In Fig.2, which white adipose tissue depot was used? gWAT? iWAT?

5. Fig.3. Similar to the case of Fig.2, tdTomato-derived fluorescence in some images is very weak and blurry (Fig.3A and 3B).

6. Fig.3. From the images of 3C, 3D, it looks like a very limited number of adipocytes are tdTomato positive (Meflin-expressing lineage cells). However, in Fig 3E and 3F, approximately half of the adipocytes are GFP positive, suggesting that these cells are Meflin-expressing lineage cells. The authors should explain this discrepancy.

7. L238, the author mentioned “Meflin-expressing lineage cells differentiate into mature adipocytes that constitute CLS”. However, they also showed that tdTomato-expressing cells in CLS also express F4/80, suggesting that Meflin-expressing lineage cells in CLS are macrophages. Therefore, the authors should clarify whether Meflin-expressing lineage cells in CLS are adipocytes or macrophages.

8. L261, “Fig 3C” should be “Fig 3D”.

9. Fig.4B. Similar to the case of Fig.2, tdTomato-derived fluorescence is too weak and blurry to judge whether these cells are tdTomato positive.

10. Fig.4. There is only one beige adipocyte in these images, and it is difficult to judge whether all beige adipocytes are derived from Meflin-expressing lineage cells. Thus, the authors should show images containing many beige adipocytes at a low magnification.

6. PLOS authors have the option to publish the peer review history of their article (what does this mean?). If published, this will include your full peer review and any attached files.

Reviewer #1: No

Reviewer #2: No

---

## [Author Response · Author response to Decision Letter 0]

23 Feb 2021

Responses to Reviewers’ comments

To Reviewer #1:

1. To what degree did Meflin mRNA and protein expression levels increase in the adipose tissue of Meflin transgenic mice compared to those of control mice?

Response:

Thank you for your insightful question. Meflin-CreERT2 mice were generated by the microinjection of a BAC-derived targeting vector into the pronuclei of fertilized eggs. For the targeting vector, CreERT2 was inserted just below the ATG of Meflin to avoid the expression of foreign DNA-derived Meflin. Southern blotting data (Fig 1B) showed that the transgenic vector did not disrupt the endogenous Meflin gene. In fact, quantitative real-time PCR revealed no difference in the expression of Meflin in the white adipose tissue of mice with or without the transgenic allele (S1 Fig A). 

We have added new data to S1 Fig A to show the profile of the Meflin gene in the mice. We have also added the following sentence to the text: “Meflin gene expression showed no difference in the adipose tissue in the presence or absence of the transgenic allele (S1 Fig A)” (Lines 203-205). We have also added two sections describing the quantitative real-time PCR method and the statistical analysis to the Experimental Methods section (Lines 172-189).

2. Do other mesenchymal stem cell (MSC) markers, such as CD90, CD105 and Sca-1 elevate in the adipose tissues of high fat diet-fed mice or after the cold exposure?

Response: 

An HFD induced the expression of MSC markers including Meflin, Cd105, and Sca-1 in mice. This result may be due to the increased proliferation of adipocyte progenitors and the subsequent differentiation of preadipocytes into mature adipocytes. In contrast, cold stimulation at 6℃ for 48 h did not affect the expression of MSC markers. Thus, we have added the following sentences to the manuscript: “The expression of mesenchymal stem cell (MSC) markers (including Meflin, Cd105 and Sca-1) was increased in the high-fat diet-loaded mice, compared with the normal diet-loaded mice (S1 Fig B)” (Lines 254-256). “Cold stimulation did not affect the expression of MSC marker genes including Meflin, Cd105, and Sca-1 (S1 Fig C)” (Lines 311-313).

3. Please show the ratio of Meflin-positive cells in the adipose tissue of normal chow-fed mice in Figure 3F.

Response: 

Thank you for this valuable suggestion. In Figure 3F, we counted and described the numbers of tdTomato-positive and EGFP-positive cells in the white adipose tissue of MeflinCreERT2 *mTmG mice after 8 weeks of normal diet (Fig 3F). The following statement has been added: “The normal diet-fed mice had an EGFP-positive rate of 26.7% and a tdTomato-positive rate of 73.3% (Fig 3F, G)” (Lines 274–275).

4. It is recommended that the authors measure the expression levels of Meflin in UCP-1 positive cells of the adipose tissue after cold stimulation.

Response: 

As this reviewer suggested, we tried to purify UCP1-positive beige adipocytes from mice under cold stimulation according to the method described in a paper by Hagberg.1 However, we were unable to purify these cells because of limitations in the equipment that were used. Our data suggested that Meflin-positive adipocyte progenitors can differentiate into not only white adipocytes, but also UCP1-positive beige adipocytes. Meflin expression is reportedly limited to mesenchymal stem cell-like cells including adipocyte progenitors and disappears once they differentiate into mature adipocytes.2 Consistent with this report,3 a recent report on single-cell RNA profiling revealed that Meflin-positive Pi16+ progenitors did not express Ucp1.

1. Hagberg CE, et al. Flow Cytometry of Mouse and Human Adipocytes for the Analysis of Browning and Cellular Heterogeneity. Cell Rep. 2018; 24(10):2746-2756.e5.

2. Maeda K, et al. Identification of Meflin as a Potential Marker for Mesenchymal Stromal Cells. Sci Rep. 2016; 6:22288.

3. Henriques F et al. Single-Cell RNA Profiling Reveals Adipocyte to Single-Cell RNA Profiling Reveals Adipocyte to Macrophage Signaling Sufficient to Enhance Thermogenesis. Cell Rep. 2020 Aug 4;32(5):107998

5. There was no description of Figure 4B’ in Result section.

Response: 

We apologize for the lack of a description. We have corrected Line 292 as follows: “To determine the cellular characteristics, we stained iWAT with anti-UCP1 and anti-tdTomato antibodies and observed that cells with a beige-like adipocyte morphology were co-stained (Fig. 4A, 4A’, 4B, 4B’)” (Lines 312-315).

6. The phenotypes of high-fat diet-fed Meflin transgenic mice, such as inflammation and insulin sensitivity, are not mentioned in the text. Please specify those phenotypes.

Response: 

Thank you for your valuable insight. Data on glucose tolerance and inflammation in the adipose tissue of Meflin transgenic mice fed a high-fat diet has been added to S1 Fig E, F. In addition, we have added the following statement at Lines 263–265: “Inflammatory markers were elevated in mice fed a high-fat diet for 8 weeks, compared with mice fed a normal diet (S1 Fig D). The transgenic expression of Meflin-CreERT2 did not affect glucose tolerance (S1 Fig E, F).” We have also added sentences describing the glucose tolerance test and the insulin tolerance test used in this study to the Method section (Lines 166-170).

To Reviewer #2

1. L120, “2 min at 4 °C” should be “2 min at 94 °C”.:

Response: Thank you very much for pointing out this error. Line 120 has been corrected to “2 min at 94℃” (Line 125).

2. The authors should determine the copy number of inserted DNA in mice used this study (Fig.2 ~) by Southern blotting.

Response:

Thank you for pointing this out. Figure 1B was submitted to show the transmission of the F1 generation to the germ line using a Southern blot. However, the band of the transgenic allele appears to be fainter than that of the endogenous Meflin. This appearance may be due to an incomplete restriction enzyme treatment, which left some residues. We have replaced the results of the Southern blot experiment confirming that the founders carried the transgenes (Fig 1B). The lower and upper bands were from the endogenous Meflin allele and the transgenic allele, respectively. A semi-quantitative analysis using ImageJ showed that the intensity of the upper band was half that of the lower band, indicating that one copy of the transgene allele had been inserted.

Lines 183–185 have been revised as followings: “A Southern blot analysis confirmed that the four founder mice carried the CreERT2 allele. The transgenic and wild-type alleles were identified at 4.9 and 2.9 kb, respectively. A semi-quantitative analysis using ImageJ showed that the intensity of the 4.9 kb band was half that of the 2.9 kb band, indicating that one copy of the transgene allele had been inserted (Fig 1B)” (Lines 198-202).

In the Generation of BAC Meflin-CreERT2 transgenic mouse strains section, we have added a description of how to semi-quantify the intensity of the bands using ImageJ (Lines 110-115).

3. Fig.2. tdTomato-derived fluorescence in some images is very weak and blurry, and it is very difficult to judge which cells are tdTomato positive cells. Especially, the fluorescence from brown adipose tissue, liver, intestine, testis, and kidney looks very weak, and it is difficult to exclude the possibility that these fluorescences are autofluorescence. The authors should show clearer images and clarify that these fluorescencea are not from autofluorescence.

Response:

We are sorry that the fluorescent photos were blurry and difficult to see. To make the images clearer, we have changed the images to ones that convey our report’s findings more clearly, paying attention to the image resolution and replacing the images with larger versions. If the PDF file created by this revision process also appears blurry, please refer to the “Click here to access~” link in the upper right corner of the PDF file to see the quality of the image submitted to the publisher. This link will take you to the images that have been submitted to the publisher.

The mentioned images of brown adipose tissue, liver, intestines, testes, and kidneys have been replaced with images that show the fluorescence more clearly (Fig 2). The description of the scale bar has been changed because it the original photos have been replaced with larger images. Line 211 has also been changed as follows: “The scale bars represent 50 μm in (B), (B’), (C), (C’), (E), (E’), (I) and (I’) and 100 μm in all the other panels” (Lines 246-248).

4. In Fig.2, which white adipose tissue depot was used? gWAT? iWAT?

Response:

Thank you for pointing this out. gWAT was used in Fig 2. We have added the following sentence: “The tdTomato signal (red) was found in gonadal white (A) and brown (B) adipocytes” (Line 242).

5. Fig.3. Similar to the case of Fig.2, tdTomato-derived fluorescence in some images is very weak and blurry (Fig 3A and 3B).

Response:

We have replaced the image with a clearer version (Fig. 3A and 3B). In Fig. 3A, we have changed the image to show the membranous structure that captures gWAT and the cells that enter the adipose tissue from the capsule surrounding gWAT more clearly. Figure 3B is a highly magnified image of Fig. 3A. For this reason, the description of Fig. 3B at Line 225 has been deleted.

6. Fig.3. From the images of 3C, 3D, it looks like a very limited number of adipocytes are tdTomato positive (Meflin-expressing lineage cells). However, in Fig 3E and 3F, approximately half of the adipocytes are GFP positive, suggesting that these cells are Meflin-expressing lineage cells. The authors should explain this discrepancy.

Response：

Thank you for your comment. The apparent discrepancy in the results shown in Figure 3C, D and E, and F is caused by a difference in the reporter mice that were used. Mature adipocytes can be detected more clearly using membrane-bound EGFP than using cytosolic tdTomato because the cytoplasmic area of mature adipocyte is relatively small. On the other hand, since the proportion of the cytoplasmic area of stromal cells is larger than that of mature adipocytes, stromal cells can be more clearly detected in tdTomato mice.

7. L238, the author mentioned “Meflin-expressing lineage cells differentiate into mature adipocytes that constitute CLS”. However, they also showed that tdTomato-expressing cells in CLS also express F4/80, suggesting that Meflin-expressing lineage cells in CLS are macrophages. Therefore, the authors should clarify whether Meflin-expressing lineage cells in CLS are adipocytes or macrophages.

Response：

Thank you for a very important question. Since it is generally accepted that macrophages are the major cell type constituting the CLS, it is very important to determine whether Meflin lineage tdTomato-positive cells at the CLS are adipocytes, macrophages, or preadipocytes. Thus, we purified tdTomato-positive cells using flow cytometry and examined the gene expressions of adiponectin, F4/80 and Pdgfra. However, the number of tdTomato cells was so small that we were unable to detect the expression of these genes. Thus, we cannot determine the identity of the tdTomato+ cells. Thus, we have corrected Line 238 as follows: “These results suggest that in adult adipose tissues in animals fed an HFD, Meflin lineage cells constitute the CLS. However, whether these cells are adipocytes, macrophages or preadipocytes remains unknown” (Lines 277-280).

8. L261, “Fig 3C” should be “Fig 3D”.

Response：

You are absolutely right. I have corrected this mistake.

9. Fig.4B. Similar to the case of Fig.2, tdTomato-derived fluorescence is too weak and blurry to judge whether these cells are tdTomato positive.

Response：

Figure 4B has been replaced with a clearer image.

10. Fig.4. There is only one beige adipocyte in these images, and it is difficult to judge whether all beige adipocytes are derived from Meflin-expressing lineage cells. Thus, the authors should show images containing many beige adipocytes at a low magnification.

Response:

Thank you for pointing this out. To show multiple beige adipocytes, the previous whole mount image shown in Fig 4A has been replaced with a low magnification image of beige adipocytes. Therefore, Lines 267-269 of the manuscript have been deleted. We have corrected Lines 278-283 as follows: “(A, A’) Immunostaining image shows tdTomato (red)- and UCP1 (green)-positive cell masses. (B, B’) Magnified images of A and A’. The cells in the center have the morphological characteristics of beige adipocytes. Scale bars represent 100 μm in (A) and (A’) and 50 μm in (B) and (B’). A, B: merged (UCP1 + tdTomato + DAPI), A’, B’:tdTomato.” (Lines 321-325).

---

## [Editor Report · Decision Letter 1]

24 Feb 2021

Generation and characterization of a Meflin-CreERT2 transgenic line for lineage tracing in white adipose tissue

PONE-D-20-35143R1

Dear Dr. Tobe,

We’re pleased to inform you that your manuscript has been judged scientifically suitable for publication and will be formally accepted for publication once it meets all outstanding technical requirements.

Kind regards,

Nobuyuki Takahashi, Ph.D.

Academic Editor

PLOS ONE

Additional Editor Comments:

Authors addressed all of the Reviewers' comments in this revised manuscript.

---

## [Editor Report · Acceptance letter]

1 Mar 2021

PONE-D-20-35143R1 

Generation and characterization of a *Meflin*-CreER^T2^ transgenic line for lineage tracing in white adipose tissue 

Dear Dr. Tobe:

I'm pleased to inform you that your manuscript has been deemed suitable for publication in PLOS ONE. Congratulations! Your manuscript is now with our production department. 

Kind regards, 

on behalf of

Dr. Nobuyuki Takahashi 

Academic Editor

PLOS ONE